# HIV-Induced Hyperactivity of Striatal Neurons Is Associated with Dysfunction of Voltage-Gated Calcium and Potassium Channels at Middle Age

**DOI:** 10.3390/membranes12080737

**Published:** 2022-07-28

**Authors:** Christina E. Khodr, Lihua Chen, Lena Al-Harthi, Xiu-Ti Hu

**Affiliations:** Department of Microbial Pathogens and Immunity, Rush University Medical Center, Cohn Research Building, Rm.610, 1735 W. Harrison Street, Chicago, IL 60612, USA; ckhodr@gmail.com (C.E.K.); lihua_chen@rush.edu (L.C.); lena_al-harthi@rush.edu (L.A.-H.)

**Keywords:** neuroHIV, aging, caudate-putamen, medium spiny neuron, hyperexcitability, electrophysiology, patch-clamping

## Abstract

Despite combination antiretroviral therapy, HIV-associated neurocognitive disorders (HAND) occur in ~50% of people living with HIV (PLWH), which are associated with dysfunction of the corticostriatal pathway. The mechanism by which HIV alters the neuronal activity in the striatum is unknown. The goal of this study is to reveal the dysfunction of striatal neurons in the context of neuroHIV during aging. Using patch-clamping electrophysiology, we evaluated the functional activity of medium spiny neurons (MSNs), including firing, Ca^2+^ spikes mediated by voltage-gated Ca^2+^ channels (VGCCs), and K^+^ channel-mediated membrane excitability, in brain slices containing the dorsal striatum (a.k.a. the caudate-putamen) from 12-month-old (12mo) HIV-1 transgenic (HIV-1 Tg) rats. We also assessed the protein expression of voltage-gated Ca_v_1.2/Ca_v_1.3 L-type Ca^2+^ channels (L-channels), NMDA receptors (NMDAR, NR2B subunit), and GABA_A_ receptors (GABA_A_Rs, β_2,3_ subunit) in the striatum. We found that MSNs had significantly increased firing in 12mo HIV-1 Tg rats compared to age-matched non-Tg control rats. Unexpectedly, Ca^2+^ spikes were significantly reduced, while K_v_ channel activity was increased, in MSNs of HIV-1 Tg rats compared to non-Tg ones. The reduced Ca^2+^ spikes were associated with an abnormally increased expression of a shorter, less functional Ca_v_1.2 L-channel form, while there was no significant change in the expression of NR2Bs or GABA_A_Rs. Collectively, the present study initially reveals neuroHIV-induced dysfunction of striatal MSNs in 12mo-old (middle) rats, which is uncoupled from VGCC upregulation and reduced K_v_ activity (that we previously identified in younger HIV-1 Tg rats). Notably, such striatal dysfunction is also associated with HIV-induced hyperactivity/neurotoxicity of glutamatergic pyramidal neurons in the medial prefrontal cortex (mPFC) that send excitatory input to the striatum (demonstrated in our previous studies). Whether such MSN dysfunction is mediated by alterations in the functional activity instead of the expression of NR2b/GABA_A_R (or other subtypes) requires further investigation.

## 1. Introduction

The success of combination antiretroviral therapy (cART) has transformed HIV/AIDS into a manageable chronic disease with HIV-Associated Neurocognitive Disorders (HAND) [1,2]. HAND are found in up to 50% of people living with HIV/AIDS (PLWH), and are expected to become more prevalent as the PLWH population ages [3,4]. However, the cause of HAND is not fully understood. A better understanding of the underlying mechanism(s) by which HIV induces neurological and neuropsychiatric dysfunction is desperately needed for developing effective treatment for HAND.

Certain key brain regions appear to be more susceptible and vulnerable to HIV-1 infection, including, but not limited to, the hippocampus, medial prefrontal cortex (mPFC), and striatum (caudate-putamen and nucleus accumbens, a.k.a the dorsal striatum and ventral striatum, respectively) [5,6,7,8]. These brain regions play critical roles in regulating neurocognition, but their function is profoundly disrupted in HAND [5,9,10]. The striatum predominantly consists of γ-aminobutyric acid (GABA)-containing medium spiny neurons (MSNs), which comprise up to 95% of neurons in this region [11]. Glutamatergic afferents from the frontal cortex (including mPFC) and thalamus provide the major excitatory driving force for MSN excitability [11,12]. Although HIV-1 does not infect neurons, toxic effects from various HIV-1 proteins and inflammatory molecules, as well as dysregulated astrocyte/neuronal interaction, could lead to neuronal dysfunction through diverse pathogenic mechanisms including excitotoxicity induced by neuronal Ca^2+^ dysregulation [13,14,15].

HIV-1 infection is found to be associated with hyperactivity of the *N*-methyl-D-aspartate receptor (NMDAR), an ionotropic glutamatergic receptor that is highly permeable to Ca^2+^ [13,16]. However, the efforts of blocking hyperactive NMDARs to alleviate severe HAND have failed in clinical trial studies [17], indicating that other mechanism(s), in conjunction with NMDAR overactivation, could jointly drive HIV-induced neuronal injury [6,13].

We have been examining how HIV-1 alters the function of voltage-gated Ca^2+^ channels (VGCCs), especially the L-type VGCCs (L-channels), and how this VGCC dysfunction contributes to HIV-induced neuronal hyperactivity, and ultimately neurotoxicity mediated by excessive neuronal [Ca^2+^]_in_. Using the HIV-1 Tg rat [18] and toxic HIV-1 protein trans-activator of transcription (Tat)-treated rats as animal models of neuroHIV, we demonstrate that HIV-1 abnormally increases the excitability of mPFC pyramidal neurons, driving them from an initial overactivation to ultimate inactivation (losing firing) [19,20,21,22,23]. We have also found that age and the duration of HIV exposure determine the impact of dysfunctional VGCCs/L-channels and K^+^ channels on this mPFC neuronal hyperexcitability. Specifically, VGCC/L-channel overactivation contributes to the mPFC neuronal hyperexcitability in adolescent (6–7 weeks-old, and young adult (6 month-[m]-old) HIV-1 Tg rats [19,22], but not in older (12mo-old) HIV-1 Tg rats [20], associated with altered mRNA expression of voltage-gated K^+^ channels (VGKCs, K_v_) [23].

Here, we used 12mo HIV-1 Tg rats to assess if and how the striatal neuronal activity and VGCC (especially L-channels)/K_v_ channel function are altered in the context of neuroHIV. As a non-infectious rodent model of neuroHIV that expresses seven of the nine HIV-1 genes (with gag and pol-deleted) [18] and exhibits certain clinical characteristics of HAND [18,24], the HIV-1 Tg rat partially models the CNS HIV reservoir and persistent inflammation in the brain despite suppression of HIV replication by cART [19,20,22,23,25].

In the current study, we initially determined neuroHIV-induced dysfunction in striatal MSNs by defining the alterations in evoked firing, Ca^2+^ spikes (reflecting activity of functional VGCCs), VGKC activity, and membrane properties. We also assessed changes in the expression of L-channels (both Ca_v_1.2 α1c and Ca_v_1.3 α1d subtypes), NMDARs, and GABA receptors (initially focusing on NR2B and GABA_A_Rs, respectively) in the caudate-putamen of 12mo-old HIV-1 Tg rats. We found that striatal MSNs display abnormally-increased firing, but significantly-reduced Ca^2+^ influx via VGCCs and enhanced VGKC activity, in 12mo-old HIV-1 Tg rats compared to age-matched non-Tg controls. Such increased MSN firing is likely due to enhanced excitatory glutamate inputs from hyperactive mPFC pyramidal neurons (that we have previously demonstrated in the context of neuroHIV [19,20,22,23]), increased activity of functional NMDARs (without overexpression), and/or reduced GABA_A_R activity (without down-regulation) in the striatum. Further investigation is needed to elucidate this potential mechanism.

## 2. Materials and Methods

*Animals.* Young male HIV-1 Tg and non-Tg F344 rats at age of 12mo (equivalent to 30-yr-old humans) [26] were housed in polycarbonate cages on a 12 h light/dark cycle with food and water available *ad libitum* at the Rush University Comparative Research Center. Rats were housed four to a cage when they weighed <200 g and two to a cage when weighed ≥200 g. Animal care and use procedures were conducted in accordance with NIH, USDA and institutional guidelines, and approved by the Institutional Animal Care and Use Committee at Rush University Medical Center. All studies were performed on 12mo-old rats (~250–300 g).

*Tissue Preparation.* Rats were anesthetized and euthanized for experiments at 12mo of age. Prior to transcardial perfusion, deep sedation was ensured by testing responsiveness to hind limb toe pinch and eye pokes. If any response was observed, additional dose was administered. Under deep sedation, the chest of a rat was cut open, and the blood vessels leading to the lower part of the body were blocked. Rats were transcardially perfused with ice-cold cutting solution (in mM: 248 sucrose, 2.9 KCl, 2 MgSO_4_, 1.25 NaH_2_PO_4_, 26 NaHCO_3_, 0.1 CaCl_2_, 10 glucose, 3 kynurenic acid, 1 ascorbic acid; pH = 7.4–7.5, 60 mL) to clear the brain tissue and to minimize the damage of neurons in the brain.

For electrophysiology, rats (5–10/group) were transcardially perfused with cutting solution (in mM: 248 sucrose, 2.9 KCl, 2 MgSO_4_, 1.25 NaH_2_PO_4_, 26 NaHCO_3_, 0.1 CaCl_2_, 10 glucose, 3 kynurenic acid, 1 ascorbic acid; pH = 7.4–7.45), and brains were coronally sectioned (300 µm). Slices were incubated in oxygenated artificial cerebrospinal fluid (aCSF; in mM: 125 NaCl, 2.5 KCl, 25 NaHCO_3_, 1.25 NaH_2_PO_4_, 1 MgCl_2_, 2 CaCl_2_, 15 glucose; pH = 7.35–7.4, 305–315 mOsm) for 1 h at room temperature before patch-clamp recording.

For biochemistry, rats (7–8/group) were transcardially perfused with 0.9% saline, and the striatum was dissected and stored at −80 °C until homogenization. Tissue was homogenized in homogenization buffer (25 mM HEPES, 500 mM NaCl, 2 mM EDTA, 0.1% Triton-Tx, 1 mM DTT) containing protease inhibitors (cOmplete™, Mini protease inhibitor cocktail, Sigma-Aldrich, St. Louis, MO, USA) and phosphatase inhibitors (PhosSTOP, Sigma-Aldrich, St. Louis, USA) using a tissue homogenizer. Homogenates were centrifuged at 14,000 rpm for 15 min at 4 °C and the supernatant was collected for western blots.

*Whole-cell patch-clamp recording*. Current-clamp recordings were conducted as previously described [19,22]. MSNs in the dorsolateral caudate-putamen were identified using differential interference contrast microscopy on a Nikon Eclipse E600FN microscope (Nikon Instruments Inc., Melville, NY, USA), and patched with recording electrodes (4–6 MΩ) filled with internal solution (pH = 7.3–7.35; 280–285 mOsm; for evoked action potentials [in mM]: 120 K-glucontate, 10 HEPES, 0.1 EGTA, 20 KCl, 2 MgCl_2_, 3 Na_2_ATP, and 0.3 NaGTP, and for voltage-gated Ca^2+^ potentials [in mM]: 140 Cs-gluconate, 10 HEPES, 2 MgCl_2_, 3 Na_2_ATP and 0.3 NaGTP).

To assess action potentials, depolarizing currents (0–400 pA) were applied for 500 ms at 25 pA intervals. Criteria for data analysis of MSNs in non-Tg rats included (1) a stable RMP more hyperpolarized than −60 mV, and (2) a rheobase (minimal depolarizing current)-evoked, initial action potential with amplitude greater than 60 mV. To assess VGCC activity (indicated by Ca^2+^ spikes), brain slices were perfused with aCSF containing selective blockers/antagonists for Na^+^ channels (0.5 µM tetrodotoxin), K^+^ channels (20 mM extracellular tetraethylammonium, 140 mM internal Cs-gluconate), NMDAR/AMPAR (2.5 mM kynurenic acid) and GABA_A_R (100 µM picrotoxin). *V*_m_ was held at ~−80 mV (average RMP) because blocking K^+^ channels causes RMP depolarization [27]. Ca^2+^ spikes were elicited using rheobase with a 40 ms duration. Criteria for data analysis included (1) consistent rheobase-evoked Ca^2+^ spikes, and (2) perfusion with aCSF containing ion channel blockers/receptor antagonists for at least 10 min. To assess *V*_m_ changes mediated by K_v_ channels, brain slices were perfused with aCSF containing selective blockers for Na^+^ (tetrodotoxin, 1 µM), Ca^2+^ (cadmium, Cd^2+^, 200 µM) channels, NMDAR/AMPA receptors (kynurenic acid, 3 mM), and GABA_A_R (picrotoxin, 100 μM).

*Protein levels of L-channels, NMDAR_2B_, and GABA_A_R_β2,3_*. Total protein (10 µg per sample) was separated by SDS-PAGE, transferred to PVDF membranes (Millipore, Billerica, MA), blocked with 5% milk or 5% BSA at RT, and incubated overnight at 4 °C in primary antibody (1:1000 rabbit anti-Ca_v_1.2, 1:1000 rabbit anti-Ca_v_1.3, Alomone Labs, Jerusalem, Israel; 1:2000 rabbit anti-NMDAR2B, Millipore; 1:1000 mouse anti-GABARβ2,3, Millipore; 1:7500 rabbit anti-actin, Santa Cruz, CA, USA). Membranes were rinsed, incubated in secondary antibody at room temperature for 1 h (1:5000 horseradish peroxidase (HRP)-conjugated goat anti-rabbit, Millipore, or HRP-goat anti-mouse, SeraCare, Milford, MA, USA), and then developed using Supersignal™ West Pico Chemiluminescent Substrate (Thermo Fisher Scientific, Waltham, MA, USA). Films were scanned as jpg images for densitometry.

*Statistical Analysis*. All data were analyzed using *t*-tests or two-way repeated measures (rm) ANOVA followed by Sidak’s post-hoc tests. Two-way rmANOVA was used to compare responses of striatal neurons at different current steps (e.g., action potential spike frequency, and *V*_m_ alterations mediated by K_v_ channels, n reflects neuron numbers). Action potential properties, calcium spike properties (n reflects neuron numbers), and protein levels (n reflects the number of rats) were compared using *t*-tests. The number of animals and neurons per group assessed in this study was determined from power analysis in combination with empirical data from our previously-published studies. Data were excluded from analysis if found to be an outlier (≥2× the standard deviation from the mean) or if experimental criteria were not met (described above in the electrophysiology section). Statistical significance was set at *p* ≤ 0.05.

## 3. Results

### 3.1. Firing of Striatal MSNs Is Abnormally Increased in 12mo HIV-1 Tg Rats

We determined the excitability of MSNs in 12mo HIV-1 Tg and non-Tg rats. This age of rats is equivalent to 30-year-old humans [26]. We assessed the characteristics of action potentials and firing frequency. Age-matched non-Tg rats were used as the control for each individual assessment in the present study. The majority of striatal MSNs had a firing/spike response evoked by rheobase ≤300 picoamperes (pA, 76.5%, 13/17 neurons in 10 non-Tg rats and 74.1%, 20/27 neurons in 14 HIV-1 Tg rats). In these MSNs (Figure 1A,B), we found a significant increase in the spike number in HIV-1 Tg rats compared to non-Tg rats (two-way rmANOVA: HIV: *F*_(1,31)_ = 5.092, *p* = 0.0312; current, *F*_(15,465)_ = 115, *p*
*<* 0.0001; interaction, *F*_(15,465)_ = 3.019, *p* = 0.0001). The passive and active membrane properties were similar in striatal neurons regardless of genotype, but there was a trend of RMP depolarization in the context of neuroHIV (*p* = 0.074, Table 1A). A small portion of MSNs (~25%) from 12mo non-Tg rats or HIV-1 Tg rats (which might have been affected by aging or injured after chronic exposure to HIV in vivo), was much less responsive to *V*_m_ depolarization, and therefore, were excluded from data analysis.

### 3.2. Activity of VGCCs Is Reduced in Striatal MSNs of 12mo HIV-1 Tg Rats

To determine if and how VGCCs contribute to the increased striatal MSN firing in HIV-1 Tg rats, we assessed VGCC activity under the conditions in which Na^+^ channel-, K^+^ channel-, NMDAR/AMPAR-, and GABA_A_R-mediated responses, were blocked. Unexpectedly, we found that MSNs in HIV-1 Tg rats (n = 8 in 4 rats) had a significantly-reduced Ca^2+^ spike duration (Figure 2B; *t*_(18)_ = 2.918, *p* = 0.0092) and area (Figure 2C; *t*_(18)_ = 2.666, *p* = 0.0158; reflecting decreased Ca^2+^ influx) compared to MSNs in non-Tg rats (n = 12 in 10 rats, Figure 2A–C). Likewise, striatal MSNs in HIV-1 Tg rats required a significantly greater rheobase to evoke a Ca^2+^ spike compared to those in non-Tg rats (*t*_(18)_ = 2.303, *p* = 0.0335; Table 1B), further indicating that VGCCs in striatal MSNs of HIV-1 Tg rats were less active than those in non-Tg rats. Other Ca^2+^ spike properties were not significantly different in Tg and non-Tg rats, although there was a trend of reduced half-amplitude duration (*t*_(13)_ = 1.872, *p* = 0.0838) in striatal MSNs from HIV-1 Tg rats compared to non-Tg rats (Table 1B). Such a trend could suggest alterations in the gating mechanism (e.g., the opening, closing, and/or inactivation) of VGCCs that might be responsible for the reduced Ca^2+^ influx.

### 3.3. Expression of a Shorter and Less Functional 150 kDa Ca_v_1.2 L-Channel Form Is Significantly Increased in the Striatum of 12mo Rats in the Context of neuroHIV

There are different subtypes of VGCCs [28], including the L-channel. In the brain, L-channels consist of a high voltage-activated (HVA-Ca_v_1.2 α1c) and a low voltage-activated (LVA-Ca_v_1.3 α1d) form [29]. We demonstrate that L-channel function and expression are abnormally increased in mPFC of HIV-1 Tg rats; associated with mPFC neuronal hyperactivity [19,20,22]. Here, we assessed L-channel protein levels in the caudate-putamen (Figure 2). We found that the levels of full-length (~250 kDa) HVA-Ca_v_1.2 and LVA-Ca_v_1.3 were not statistically different in the caudate-putamen of Tg (n = 8) and non-Tg rats (n = 7, Figure 2D–F). However, the expression of a shorter, and likely less functional [30], 150 kDa Ca_v_1.2 form was significantly higher in the caudate-putamen of HIV-1 Tg rats (n = 7) compared to non-Tg rats (n = 7; 150 kDa: *t*_(12)_ = 2.022, *p* = 0.0479; 150 kDa/250 kDa: *t*_(12)_ = 3.533, *p* = 0.0305; Figure 2D,G,H). This dysregulation also induced a significant increase in the ratio of 150 kDa vs. 250 kDa Ca_v_1.2 form (Figure 2H). These data suggest that the reduced voltage-sensitive Ca^2+^ influx among striatal MSNs in HIV-1 Tg rats results from an abnormal increase of less functional L-channels due to increased expression of the shorter, 150 kDa Ca_v_1.2 form.

### 3.4. Expression of NMDAR and GABA_A_R Is Unaltered in the Striatum of 12mo Rats by neuroHIV

Our data show that Ca^2+^ influx was reduced in striatal MSNs of HIV-1 Tg rats but under blockade of NMDARs and GABA_A_Rs. Thus, it is possible that alterations in excitatory or inhibitory synaptic activity could facilitate VGCC activity, leading to increased firing. To examine the potential influence of NMDARs and GABA_A_Rs, we assessed the protein levels of NMDAR NR2B subunits and GABA_A_R β_2,3_ subunits in the caudate-putamen (Figure 3). We chose to assess the NR2B subunit because it is functionally involved in excitotoxicity in neurons [31,32]; while we chose to assess the β_2,3_ subunit of GABA_A_R because the β_2_ and β_3_ subunits, but not β_1_, are detected in the rat striatum [33]. However, we did not find any significant difference in NR2B (n = 7 HIV-1 Tg vs. n = 8 non-Tg, Figure 3B), or GABA_A_Rβ_2,3_ (n = 8 HIV-1 Tg vs. n = 7 non-Tg, Figure 3C) protein in the caudate-putamen between HIV-1 Tg and non-Tg rats. These data suggest that striatal hyperexcitability was not mediated by increased NR2B or decreased GABA_A_R β_2,3_. But it remains to be determined if (i) NMDAR activity is increased; (ii) GABA_A_R activity is reduced, and (iii) changes in the expression of other NMDAR subunits or AMPARs exist, in 12mo HIV-1 Tg rats.

### 3.5. K_v_ Channel Activity Is Significantly Increased, While Activity of K_ir_ and K_2P_ Channels Is Not Significantly Altered, in Striatal MSNs in the Context of neuroHIV

Excitability (*V*_m_) of striatal MSNs is also mediated by various subtypes of K^+^ channels. Thus, we first assessed the activity of K_v_ channels that conduct K^+^ efflux. We found that K_v_ activity (activated by *V*_m_ depolarization) was significantly increased, indicated by enhanced downward *V*_m_ traces (Figure 4A, left) in HIV-1 Tg rats (non-Tg vs. Tg, n = 9 MSNs/7 rats vs. 11 MSNs/9 rats; HIV: *F*_(1,18)_ = 1.513, *p* = 0.234; current: *F*_(19,342)_ = 161.5, *p* < 0.001; interaction: *F*_(19,342)_ = 1.782, *p* = 0.024) (Figure 4A,B). This increased K_v_ activity could diminish VGCC activity, rather than increase, MSN firing. Then we also assessed the activity of inwardly rectifying K^+^ (K_ir_) channels, which were activated by *V*_m_ hyperpolarization to conduct K^+^ influx. There was no significant difference in K_ir_ activity between non-Tg (n = 8 MSNs/6 rats) and HIV-1 Tg rats (n = 10 MSNs/8 rats. HIV: *F*_(1,15)_ = 0.5037, *p* = 0.4881; current: *F*_(15,240)_ = 233.8, *p* < 0.001; interaction: *F*_(15,240)_ = 1.295, *p* = 0.2056, Figure 4C,D). Finally, we assessed the K_2P_ channel activity that mediates the resting membrane potential (RMP). Given that RMP was maintained and regulated predominantly by outflowing K^+^ currents through activating K_2P_ channels in neurons at the resting status, we also determined if HIV altered the RMP of striatal MSNs by disturbing the function of K_2P_ channels. We found that there was a trend in which RMP of striatal MSNs was slightly, but not significantly depolarized, in the context of neuroHIV at 12mo of age (n = 12 MSNs/8 rats) compared to age-matched non-Tg control rats (n = 9 MSNs/7 rats; non-Tg vs. HIV-1 Tg: −78.5 ± 1.3 mV vs. −75.1 ± 1.2 mV; *t*-test: *t*_19_ = 0.266, *p* = 0.074; Table 1A).

Collectively, neuroHIV-induced changes in voltage-gated membrane ion channels, as well as unaffected protein levels of ionotropic NMDARs and GABA_A_Rs among striatal MSNs in 12mo rats are summarized in Table 2.

## 4. Discussion

In the present study, we initially examined how neuroHIV disrupts striatal MSN activity to define potential mechanisms that could be targeted for the treatment of HAND. We demonstrate that neuroHIV abnormally increases MSN excitability in 12mo-old HIV-1 Tg rats compared to age-matched non-Tg rats. Surprisingly, this increase was associated with reduced VGCC function and enhanced VGKC activity, rather than the opposite conditions expected. Concurrently, we also found that the protein expression of a shorter, and less functional form of Ca_v_1.2 L-channels was significantly increased in the dorsal striatum of HIV-1 Tg rats, while the protein expression of NMDAR (NR2B subunit) and GABA_A_R (β2,3 subunits) was unchanged. These novel findings indicate that the hyperexcitability of striatal MSNs in the context of neuroHIV cannot be attributed to VGCC upregulation or reduced VGKC activity (found in younger HIV-1 Tg rats); but suggest that it could be mediated by increased activity of NMDARs (e.g., opening probability/frequency/time without overexpression), in response to enhanced glutamate input from the hyperactive mPFC [19,20,22,23], reduced glutamate uptake by astrocytes, or decreased GABA_A_R activity in the HIV-altered striatum. Future study is needed to define this potential mechanism during aging.

One major finding of the current study is that MSNs have significantly-increased firing in HIV-1 Tg rats. To our knowledge, this is the first study in the field that reveals hyperexcitability of striatal MSNs from 12mo-old rats in the context of neuroHIV. The characteristics of action potentials (Table 1A) that provide a glimpse into the function of membrane ion channels that are critically involved in generating firing suggest that the mechanism underlying this MSN hyperactivity does not include increased function of voltage-gated Na^+^/Ca^2+^ channels, or reduced K_v_ channel activity. Moreover, the unchanged resting membrane potential, input resistance, afterhyperpolarization, and inward rectification, which were mediated by K_2P_ channels, Ca^2+^-activated K^+^ channels, and K_ir_ channels, respectively (Table 1A), also exclude their role in causing such MSN hyperactivity in 12mo-old rats in the context of neuroHIV.

In the current study, we also found that VGCC function was uncoupled from abnormally-increased firing in the context of neuroHIV. In fact, VGCC activity was reduced (instead of increased) in striatal MSNs of 12mo-old HIV-1 Tg rats, which was the opposite of an expectation if VGCC overactivation contributed to this MSN hyperexcitability. This HIV-induced reduction of VGCC function in striatal MSNs differs remarkably from what we previously demonstrated in mPFC pyramidal neurons [20], which could be a compensatory response to dampen the impact of excessive excitatory inputs from the hyperactive mPFC, where the functional activity of glutamatergic pyramidal neurons is significantly increased by HIV [19,20,22], or a consequence of overactivation-caused VGCC inactivation in the context of neuroHIV [22,34,35]. Thus, further study is also needed to determine if this seemingly compensatory reduction, or over-activation-induced inactivation in Ca^2+^ influx mediated by dysfunctional Ca_v_1.2 L-channels (see below), is related to cognitive deficits induced by HIV, and if so, whether such Ca^2+^ dysregulation can be intervened pharmacologically.

VGCCs are known to undergo activity-dependent regulation [28,36], which is crucial because disturbed neuronal Ca^2+^ homeostasis can cause dysregulation, damage, and even death of neurons, by disturbing excitability, signal transduction, and gene expression pathways [6,37,38]. Different forms of activity-dependent regulation have been identified, which operate over a range of time frames [28,36]. Such regulatory processes include proteolytic cleavage [30,36,39] and channel internalization [36,40]. Here, we found the significantly-increased expression of a shorter (150 kDa) form of the Ca_v_1.2 L-channel in the caudate-putamen of HIV-1 Tg rats. Expression of this shorter 150 kDa Ca_v_1.2 form could result from cleavage of the full-length L-channel. In fact, proteolytic processing of full-length Ca_v_1.2 to a 150 kDa cleavage product is reported to produce partially (i.e., less) functional L-channels, with distinct biophysical characteristics that alter the properties of full-length L-channels [30]. This is in line with our findings of reduced Ca^2+^ influx via VGCCs. It is also possible that the shorter 150 kDa Ca_v_1.2 form found here results from increased expression of an alternatively-spliced Ca_v_1.2 channel. Further studies are needed to determine whether this 150 kDa Ca_v_1.2 form is a product of proteolytic processing or alternative splicing.

Other forms of activity-dependent negative regulation, which could also contribute to the reduced VGCC function, include channel internalization [40], the formation of a potent autoinhibitor produced by proteolytic C-terminal cleavage [39], and/or calmodulin-based/Ca^2+^-dependent inactivation [28,36]. Finally, further VGCC regulation by VGCC auxiliary subunits (α_2_δ, β, and γ subunits), which modulate functional expression and trafficking of VGCCs to the membrane surface of neurons, could also contribute to the reduced Ca^2+^ influx [41,42]. Whether activity and/or expression of these L-channel subunits are altered in the context of neuroHIV requires further investigation.

It is also important to note that VGCCs consist of a number of different channel subtypes [28]. Although we have focused on L-channels, other VGCCs, including the P/Q-, N-, R-, and/or T-type Ca^2+^ channels [28] could also contribute to the reduced Ca^2+^ influx found in HIV-affected striatal MSNs. For instance, HIV alters the function of P/Q-channels in the nucleus accumbens [43] and suprachiasmatic nucleus [44], and N-channels are involved in neuropathic pain [45], a symptom that is prevalent in HIV. Additionally, findings from our previously studies also suggest that HIV alters the function/expression of LVA-Ca^2+^ channels (i.e., T-type and/or Ca_v_1.3, L-type) [19,20,22,23]. The impact of neuroHIV on neuronal Ca^2+^ dysregulation due to alterations in non-L-channels also is an interesting topic for future studies.

Alterations in excitatory (and/or inhibitory) neurotransmission are the likely cause of this MSN hyperexcitability in 12mo HIV-1 Tg rats. They could stem from differing sources including dysregulation of presynaptic activity, local astrocyte activity, and NMDAR or GABA_A_R expression and/or function in striatal MSNs. We previously demonstrated that mPFC pyramidal neurons, which project excitatory inputs to neurons in the caudate-putamen [12], are hyperactive in the context of neuroAIDS modeled in HIV-1 Tg rats regardless of age [19,20,22]. Enhanced glutamate input from which could aberrantly enhance MSN excitability by promoting the activity of NMDARs without overexpression. Additionally, HIV impairs glutamate uptake by astrocytes via disrupting β-catenin signaling, increasing local levels of extracellular glutamate [15,46,47], and therefore, facilitating neuron firing. HIV also downregulates GABAergic systems [48,49], which could also enhance striatal MSN excitability. Finally, although NR2B upregulation and/or GABA_A_R downregulation could be additional mechanisms of altered neurotransmission, our current data exclude the likelihood of their involvement. However, because here we only initially evaluated the protein expression of the NR2B subunit of NMDARs and the β2,3 subunit of GABA_A_Rs, further study is needed to determine if the activity of NR2Bs and GABA_A_Rs; specifically, their opening probability, opening frequency, and/or opening time, or other subtypes of glutamate and GABA receptors, play a key role in this MSN dysfunction. Together, the present study suggests that the functional activity of functional NMDARs is increased in response to enhanced glutamate inputs from the mPFC, and that may also be associated with reduced reuptake of glutamate by astrocytes in the striatum. Further, the activity of functional GABA_A_R could also be reduced, without changes in their expression.

Regardless of the exact source of enhanced excitatory neurotransmission that increases striatal MSN excitability in the context of neuroHIV, sustained NMDAR activation has been shown to result in internalization and degradation of Ca_v_1.2 L-channels in rat cortical neurons [50], as well as reduced Ca^2+^ currents via L-channels in rat ventricular myocytes [51]. Our novel findings suggest that chronically-enhanced excitatory neurotransmission initiates negative feedback to alter the function and expression of Ca_v_1.2 L-channels (and likely other non-L-type VGCCs). Continued glutamate stimulation could also induce the internalization of glutamate receptors [52,53,54]. In combination with L-channel internalization [50], they could ultimately reduce Ca^2+^ influx to protect neurons from excitotoxicity-induced injury or death [6,28,36,37,50], at least for a limited period of time. Accordingly, we did not observe any evident neuron death in the caudate-putamen of HIV-1 Tg rats based on NeuN protein assessments. Further studies that assess these processes in older HIV-1 Tg rats (≥24mo) would be informative regarding HIV-induced excitotoxicity during aging.

In summary, our novel findings from this study initially reveal striatal neuronal dysfunction in the mesocorticostriatal pathway of aging rats in the context of neuroHIV, which is crucial in regulating neuronal/astrocytic function and neurocognition disturbed by HIV. Together, these findings, in combination with our previous studies, suggest that HIV-induced mPFC neuronal overactivation causes abnormally-enhanced glutamate output, which induces hyperactivity of striatal MSNs, and which is associated with dysfunction of L-channels and K_v_ channels. However, these findings unexpectedly differ from VGCC upregulation and reduced VGKC activity we found in glutamatergic mPFC pyramidal neurons of youth rats in the context of neuroHIV. Further, because rat age at 12mo is equivalent to humans at middle age (~30-yr-old) [26], such neuronal and L-channel/K_v_ channel dysfunction could be exacerbated during further aging. Advancing our understanding of how HIV-1 disrupts the most vulnerable and susceptible brain regions that regulate neurocognition, and whether chronic exposure to HIV induces channelopathies in the brain, will aid in defining therapeutic targets for HAND.

## Figures and Tables

**Figure 1 membranes-12-00737-f001:**
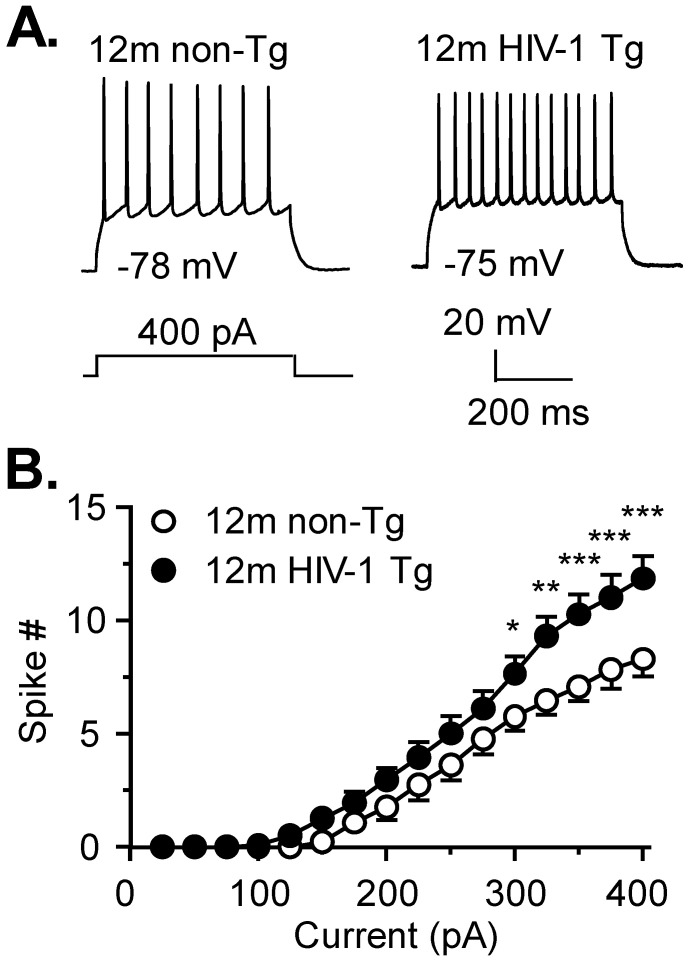
**Evoked firing is increased in HIV-exposed striatal MSNs of 12mo rats.** (**A**) Representative traces showing evoked firing (at 200, 300 and 400 pA) from striatal MSNs in the caudate-putamen of a 12mo-old non-Tg (**left**) and age-matched HIV-1 Tg (**right**) rat. (**B**) The number of action potentials elicited by each depolarizing current pulse step is graphed for non-Tg (open circles, n = 13 neurons in 7 rats) and HIV-1 Tg rats (filled circles, n = 20 neurons in 13 rats) showing that striatal MSNs from HIV-1 Tg rats are hyperexcitable compared to those from non-Tg rats. These data are from the majority of neurons, which displayed a spike response to rheobase ≤300 pA, from non-Tg and HIV-1 Tg rats, respectively. Significance is denoted as *: *p* ≤ 0.05, **: *p* ≤ 0.01, and ***: *p* ≤ 0.001.

**Figure 2 membranes-12-00737-f002:**
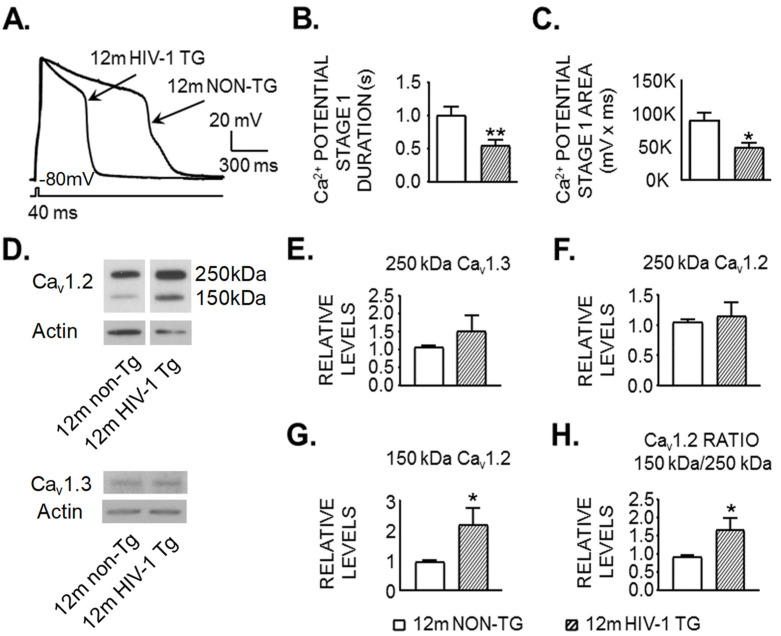
Reduced Ca^2+^ influx via VGCCs is accompanied by increased expression of a shorter form, less functional Ca_v_1.2 L-channel in the HIV-exposed dorsal striatum of 12mo rats. (**A**) Representative traces showing evoked Ca^2+^ spikes in striatal MSNs from a 12mo-old non-Tg and HIV-1 Tg rat. (**B**,**C**) The Ca^2+^ spike duration (**B**) and area (**C**), which reflect Ca^2+^ influx through VGCCs, are graphed. Note that striatal neurons from HIV-1 Tg rats (hatched bars, n = 8 in 4 rats) had significantly reduced Ca^2+^ influx compared to those from non-Tg rats (open bars, n = 12 in 10 rats). (**D**–**H**) L-channel protein levels were measured in the caudate-putamen dissected from 12mo-old HIV-1 Tg (n = 7–8) and non-Tg rats (n = 7). (**D**) Representative western blots. For the representative Ca_v_1.2 bands, the samples were run on the same gel, but were inverted compared with the order shown; i.e., the bands were rearranged to achieve the desired order for presentation. Appendix A shows the full blots for the representative samples. (**E**–**H**) Bar graphs showing relative levels of L-channel protein normalized to β-actin protein for the full length LVA-Ca_v_1.3 (**E**), the full-length (~250 kDa) HVA-Ca_v_1.2 (**F**), the shorter, 150 kDa form of Ca_v_1.2 (G), and the Ca_v_1.2 ratio, 150 kDa/250 kDa (**H**). Significance is denoted as *: *p* ≤ 0.05, **: *p* ≤ 0.01; K = 1000.

**Figure 3 membranes-12-00737-f003:**
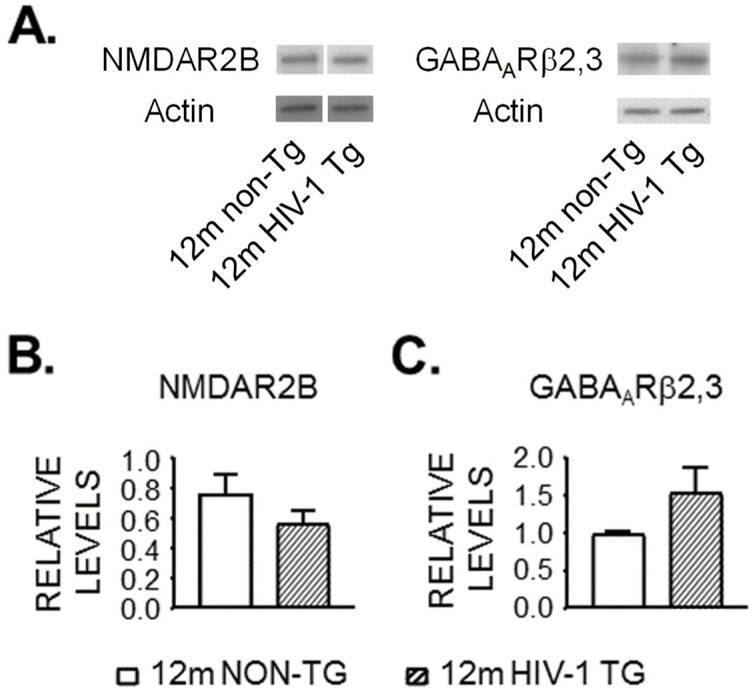
**NMDAR2B and GABA_A_R protein levels are unaltered in the HIV-exposed striatum of aging rats.** Protein levels were measured in the caudate-putamen dissected from 12mo-old HIV-1 Tg (hatched bars, n = 7–8) and non-Tg rats (open bars, n = 7–8). (**A**) Representative western blots of NMDAR2B and GABA_A_Rβ_2,3_ protein. For GABA_A_R staining and respective actin staining, membranes were first probed for GABA_A_R protein, then stripped and re-probed for actin, For the representative NMDAR bands, the samples were run on the same gel, but were not side-by-side; i.e., the bands shown were cropped from two different areas of the same membrane. Appendix A shows the full blots for the representative samples. (**B**,**C**) Bar graphs showing relative levels of NMDAR2B (**B**) and GABA_A_Rβ_2,3_ (**C**) normalized to β-actin protein.

**Figure 4 membranes-12-00737-f004:**
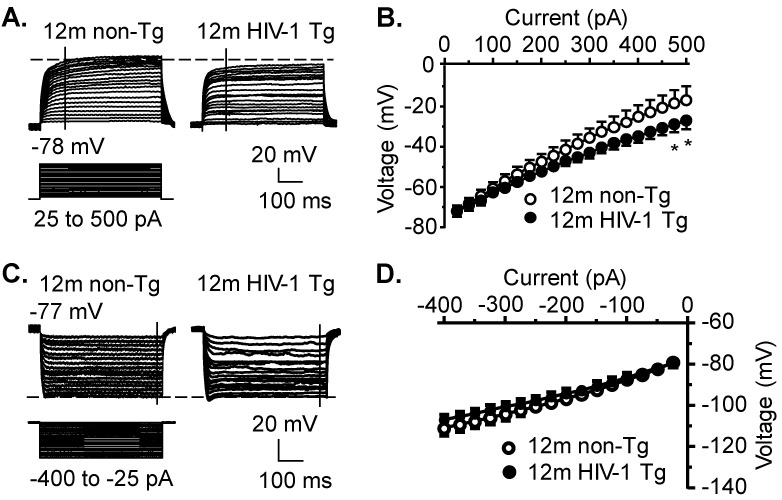
K_v_ channel activity is significantly increased, while activity of functional K_ir_ and K_2P_ channels is not significantly altered, in striatal MSNs in the context of neuroHIV. (**A**) The responses of membrane potential (*V*_m_) of striatal MSNs to excitatory stimuli (depolarization), reflecting the activation of K_v_ channels and outflowing voltage-sensitive K^+^. The vertical dash-lines indicate the time point (100 ms) at which *V*_m_ changes were assessed. (**B**) The current-voltage (*I–V*) relationships indicate a significant change in *V*_m_ mediated by K_v_ channels (12mo non-Tg: n = 11 neurons in 9 rats vs. 12mo HIV-1 Tg: n = 9 neurons in 7 rats). Significance is denoted as * *p* ≤ 0.05. (**C**) The *V*_m_ changes of striatal MSNs to inhibitory stimuli (hyperpolarization), reflecting the activation of K_ir_ channels and inwardly rectifying K^+^. The vertical dash-lines indicate the time point (450 ms) at which *V*_m_ changes were assessed. (**D**) The current-voltage (*I–V*) relationships show no significant change in *V*_m_ mediated by K_ir_ channels (12mo non-Tg: n = 10 neurons in 8 rats vs. 12mo HIV-1 Tg: n = 8 neurons in 6 rats).

**Table 1 membranes-12-00737-t001:** The impact of neuroHIV on functional activity of striatal MSNs.

*A. Membrane Property*	12mo Non-Tg	Cell #	12mo HIV-1 Tg	Cell #	*p* Value
**Rat Number**	7		14		
**RMP (mV)**	−78.5 ± 1.3	13	−75.1 ± 1.2	22	0.074
**R_in_ (MΩ)**	97.1 ± 6.7	13	108.3 ± 9.6	21	0.410
**Rheobase (pA)**	203.8 ± 12.0	13	191.7 ± 13.0	21	0.526
**Threshold (mV)**	−39.3 ± 0.8	13	−38.5 ± 1.3	22	0.671
**Spike Amplitude (mV)**	79.4 ± 2.5	13	73.0 ± 2.8	22	0.133
**1/2 Peak Duration (ms)**	2.1 ± 0.1	13	1.9 ± 0.1	22	0.371
**Time Constant (ms)**	11.5 ± 1.0	13	11.1 ± 1.1	21	0.816
**AHP (mV)**	−14.0 ± 0.6	13	−13.6 ± 0.6	21	0.659
** *B. Ca^2+^ Spike* **	**12mo Non-Tg**		**12mo HIV-1 Tg**		***p* Value**
	**Mean ± SEM**	**Cell #**	**Mean ± SEM**	**Cell #**	
**Rheobase (pA)**	555.8 ± 43.5	12	743.1 ± 75.8 *	8	0.0335
**Threshold (mV)**	−8.3 ± 2.0	8	−16.4 ± 3.4	7	0.8465
**Amplitude (mV)**	43.5 ± 2.1	8	48.2 ± 5.3	7	0.3992
**1/2 amplitude duration (ms)**	473.8 ± 91.7	8	282.3 ± 29.6	7	0.0838
**Stage 1 duration (s)**	1.01 ± 0.12	12	0.56 ± 0.07 **	8	0.0092
**Stage 1 area (KmV × ms)**	89.9 ± 11.7	12	49.2 ± 6.2 *	8	0.0158

#: number of neurons assessed, * *p* ≤ 0.05, ** *p* ≤ 0.01.

**Table 2 membranes-12-00737-t002:** Summary of the changes in membrane ion channels and ionotropic receptors assessed in GABAergic medium spiny neurons from 12mo HIV-1 rats in comparison with age-matched non-Tg rats.

Ion Channels/Ionotropic Receptors	Location	Participation in Unique Findings
1. L-type VGCC (high/low voltage-activated)	the dorsal striatum	Increased expression and ratio of dysfunctional (less active) HVA-Cav1.2 L-channel protein
2. non-L-type VGCC (high/low voltage-activated)	the dorsal striatum	Not assessed
3. Potassium channels	the dorsal striatum	Increased K_v_ channel activity; but no significant change in the activity of K_2P_ or inwardly rectifying K^+^ channels
4. NMDAR-2B and GABA_A_R	the dorsal striatum	No significant change in the protein levels of either one

## Data Availability

All the analyzed data related to this study are provided here. Further discussion and question can be directed to the corresponding author.

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
