# Peer review of "HIV-Induced Hyperactivity of Striatal Neurons Is Associated with Dysfunction of Voltage-Gated Calcium and Potassium Channels at Middle Age"

_membranes, 2022, doi:10.3390/membranes12080737_

Round 1

Reviewer 1 Report

There is so much data that the reader needs tables to separate the findings  according to ion channels, HIV animal model, role of glutamate in young vs mature animals. I hav supplied some suggestions as to how the authors may assist the readers in seeing through the mass of data. 

Reviewer 2 Report

Questions, recommendations, Comments for Authors

1)  In the legend of Figure 1, it is written that “(A) Representative traces showing evoked firing (at 200, 300 and 400 pA) from striatal MSNs...” But as I see in panel A, only 400pA was applied.

I think, you should delete the other numbers: 200, 300.

2)      In the “Results”, in line 180, 181 and 240, I do not understand what the F values mean and what the indexes mean. Please explain this.

3)      Similarly, in lines 190 and 213, what do “t” and its index mean? I am probably not enough well versed in this field, but please explain to me what they mean! Are these belonging somehow to the statistical F- and t-tests?

4)      In line 200, you wrote that: “Such trend could suggest alterations in the gating mechanism (e.g., the open probability and time) of VGCCs”. This sentence is rather superficial. For the gating mechanism of ion channels, the open probability or time is not an example. Here, it is not clear to me what is meant by time.  I think for the gating mechanism an example is the opening, or closing mechanisms, or inactivation. The open probability is not a mechanism. I suggest to rephrase this sentence.

6)      What I miss in the discussion is the explanation that I think is the most obvious. Namely: The increased firing of striatal MSNs is a straightforward consequence of the reduced activity of VGCCs and increased Kv channel activity. Because, as we all know, Kv channels are responsible for repolarization during the action potential (AP) in neurons and in muscle cells as well. Because of this, the increased Kv channel activity shortens the duration of AP. While VGCC activity is responsible for the prolongation of AP. (In neurons, there are no VGCC, this is why the AP of neurons is shorter than the AP of muscle cells). So, the reduced activity of VGCCs should cause shortening of AP.  To sum up: both phenomena shorten the AP in my opinion, i.e. they reinforce each other.

I would like to add that I am not an expert on APs. I only know the most elementary things about it (although it is true that I am about to delve into this topic, because I want to do AP measurements for one of my projects in the near future).

Please think about the explanation I have outlined and only post it in the discussion if you agree with it.

Overall, I like the style of the manuscript, as it is written. The results are clear, and I think they reinforce each other. The manuscript is well written. The quality of the figures is also good.

Author Response

Pleases see attached file.
